# Controllable Construction of Amino-Functionalized Dynamic Covalent Porous Polymers for High-Efficiency CO_2_ Capture from Flue Gas

**DOI:** 10.3390/molecules27185853

**Published:** 2022-09-09

**Authors:** Mingyue Qiu, Haonan Wu, Yi Huang, Huijuan Guo, Dan Gao, Feng Pei, Lijuan Shi, Qun Yi

**Affiliations:** 1School of Chemical Engineering and Pharmacy, Wuhan Institute of Technology, Wuhan 430205, China; 2School of Energy, Power and Mechanical Engineering, North China Electric Power University, Beijing 102206, China; 3Hubei Yihua Chemical Technology R&D Co., Ltd., Yichang 443208, China; 4Shanxi-Zheda Institute of Advanced Materials and Chemical Engineering, Taiyuan 030024, China

**Keywords:** CO_2_ capture, dynamic covalent polymer, imine exchange, amino functionalization

## Abstract

The design of high-efficiency CO_2_ adsorbents with low cost, high capacity, and easy desorption is of high significance for reducing carbon emissions, which yet remains a great challenge. This work proposes a facile construction strategy of amino-functional dynamic covalent materials for effective CO_2_ capture from flue gas. Upon the dynamic imine assembly of N-site rich motif and aldehyde-based spacers, nanospheres and hollow nanotubes with spongy pores were constructed spontaneously at room temperature. A commercial amino-functional molecule tetraethylenepentamine could be facilely introduced into the dynamic covalent materials by virtue of the dynamic nature of imine assembly, thus inducing a high CO_2_ capacity (1.27 mmol·g^−1^) from simulated flue gas at 75 °C. This dynamic imine assembly strategy endowed the dynamic covalent materials with facile preparation, low cost, excellent CO_2_ capacity, and outstanding cyclic stability, providing a mild and controllable approach for the development of competitive CO_2_ adsorbents.

## 1. Introduction

Excessive CO_2_ emissions from the burning of fossil fuels have caused a series of ecological and environmental problems [1,2]. Post-combustion capture of CO_2_ from flue gases has been considered as one of the potential strategies for reducing carbon emissions [3,4]. Various CO_2_ capture techniques, mainly including liquid ammonia absorption [5], membrane separation [6], and adsorption [7], have been widely developed. In particular, CO_2_ adsorption by solids has the advantages of high recycling rate, simple operation, low equipment corrosion, and low energy consumption; thus, it is of high promise for low-energy CO_2_ capture [8,9,10]. It is, therefore, of great significance for developing high-efficiency adsorbents with low cost, high capacity, and easy desorption, which yet remains a great challenge.

Great efforts have been made toward developing diverse kinds of CO_2_ adsorbents, e.g., inorganic porous materials, covalent organic materials, and supramolecular organic materials [11]. Inorganic materials (e.g., zeolite [12], activated carbon [13], mesoporous silica [14], and mesoporous fibers [15]), despite having the advantages of high mechanical stability and regular porosity, mainly suffer from difficult structural functionalization and intrinsic acidic sites unfavorable for CO_2_ adsorption [16]. Organic porous materials (e.g., polymers) have much more superior structural tunability compared to inorganic materials [17,18], but still suffer from complex preparation processes and high cost [19,20]. Alternatively, supramolecular organic materials, such as metal organic framework materials (MOFs) [21], hydrogen-bonded organic framework materials (HOFs) [22], covalent organic framework materials (COFs) [23], and coordination polymers [24], are generally constructed through dynamic covalent and/or noncovalent bonding interactions. The dynamic noncovalent nature endows supramolecular organic materials with remarkable features such as simple preparation process, adjustable structure, and easy functionalization. In particular, dynamic covalent bonding (DCC) combines the stability and dynamic reversibility of covalent bonding, thus allowing the resultant dynamic covalent functional materials assembled to possess facile preparation, high stability, and promising functionalization [25,26,27,28].

Amino functionalization of porous material has been considered as an efficient tool for improving CO_2_ adsorption performance by taking advantage of the interaction between the amino group and CO_2_ [29,30,31,32,33]. Traditionally, amino functionalization is mainly achieved through covalent grafting [34] and impregnation [35]. The former method often requires synthesis and activation of the as-prepared support (e.g., silica or zeolite) prior to grafting, complex chemical synthesis during the grafting process, and removal of solvent and/or unreacted reactants to activate pores [36]. The wet impregnation method tends to induce pore blockage and leaching of active component (e.g., organic amine or ionic liquid), and the liquid nature of active component increases both the viscosity of the composite material and the transfer resistance of CO_2_, thus generally inducing a relatively low CO_2_ adsorption and/or low stability [37]. Moreover, the control over the content of functional sites and the nature of textural structures, which highly affects the CO_2_ adsorption performance of the amine-functionalized materials, suffers from high challenges in both methods. Therefore, it is critical to explore a green and efficient approach for preparing amino-functionalized materials to promote CO_2_ adsorption.

Herein, this work proposes a facile construction approach of functional dynamic covalent polymers through the dynamic imine assembly of N-site rich motif and aldehyde-based spacers for high-efficiency CO_2_ capture. By virtue of the dynamic noncovalent nature, the structure of the dynamic covalent polymers can be facilely regulated through the assembly process control, thus offering a feasible approach for promoting the CO_2_ adsorption ability. Most importantly, amino-functional units can be facilely introduced through the imine exchange method, thus offering a mild and controllable approach for the development of competitive CO_2_ adsorbents.

## 2. Results and Discussion

### 2.1. Characterization of Dynamic Covalent Materials

A series of functional dynamic covalent nanomaterials were constructed via the self-assembly of 3,3′-dithiobis (propionyl hydrazine) (DTPH) and polyaldehyde at the ethyl acetate/water interface at 30 °C. Upon the self-assembly of DTPH and *p*-phthalaldehyde (PA), white powers were typically afforded (Figure 1). To verify the successful imine assembly, the resultant samples were characterized by FT-IR technology. As shown in Figure 1, the N–H vibration peaks of DTPH at 3285 and 3326 cm^−1^ disappeared in the dynamic covalent polymers [38], and a new peak assigned to C=N bond appeared at 1670 cm^−1^ [39,40], indicating that DTPH was successfully assembled with PA.

Solid-state ^13^C-NMR characterization was used to further verify imine assembly. As shown in Figure 2, the characteristic signal at 163 ppm corresponded to the chemical shift of the C=N bond in PAP−1 [41,42]. In addition, the peak of the benzene ring derived from PA at 135 ppm and the peaks of C–C derived from DTPH at 35 ppm can observed. The phenomena above proved the successful assembly of PA and DTPH through imine bonding.

SEM observation revealed that nanoflowers with spongy pores were formed in the PAP samples with various molar ratios of PA and DTPH (Figure 3a,b), and the porous structures were considered of high promise for gas adsorption.

Similarly, the self-assembly of DTPH and 4,4′-biphenyldicarboxaldehyde (BPDA) could also be achieved (Figure 2 and Figure 4). When the molar ratio of BPDA/DTPH was 1:0.5, porous nanospheres were formed (Figure 5a,b). With increasing content of DTPH, of great interest, nanospheres evolved into hollow nanotubes (Figure 5e,f). It can be noted that spongy porous structures were spread on the surface of the nanotubes, which was also beneficial for gas capture.

The thermodynamic and kinetic analysis of supramolecular polymerization is of vital importance for understanding the pathway of structural regulation [43]. The effect of reaction time on the formation of the polymers was studied by taking PAP-3 as an example. It was found that powders could be formed within a quite short time. SEM characterization verified that a regular nanoflower morphology was formed within 0.5 h (see Figure 6). It can be seen that the dynamic imine assembly of PA and DTPH was under thermodynamic control rather than kinetic control [44,45].

The crystal structures of the dynamic covalent polymers PAP and PBP were analyzed by X-ray diffraction (XRD). As can be seen from Figure 7, all the polymers had two broad diffraction peaks around 17° and 25°, indicating that the dynamic covalent polymer had an amorphous structure [46].

The maximum weight loss temperatures of PAP-1 and PBP-1 were both 292 °C, as shown in Figure 8, indicating the excellent thermal stability of the materials. The generation of imine from the reaction of amine and aldehyde is a reversible reaction that operates under thermodynamic control, through which the kinetically competing intermediate products are replaced by the thermodynamically most stable product after sufficient assembly time [47]. That is, imine assembly can afford a dynamic covalent material with excellent thermal stability under mild conditions.

Porosity is a crucial factor for gas adsorption. The porous structures of PAP and PBP samples were characterized using N_2_ adsorption-desorption. As shown in Figure 9a, the N_2_ adsorption-desorption curves of the PAP samples presented a typical H4-type hysteresis loop, reflecting that the samples were composited of mesopores and/or macropores, which could be further confirmed by the analysis of pore volume (Table 1). This result is consistent with the SEM observation. H4-type N_2_ adsorption-desorption curves with highly enlarged hysteresis loops could be observed for the PBP samples, especially for PBP-3. This phenomenon implies a relatively increased pore size in PBP-3, which was probably caused by the formation of hollow structures. It can be seen that the specific surface area of PAP-2 was higher than that of PAP-1 and PAP-3. Compared to PAP-1 and PAP-3, the molar ratio of DTPH:PA in PAP-2 was 1:1, enabling DTPH and PA to react completely to form PAP, which was helpful for a high porosity. Moreover, there was no excess branch chain from DTPH and PA blocking the pore channels, which was also helpful for a higher specific surface area of PAP-2 than that of PAP-1 and PAP-3. For PBP-3, the relatively higher specific surface area was possibly contributed by the specific hollow nanotube structures covered with spongy pores, which had much higher pore volume than PBP-1 and PBP-2.

### 2.2. CO_2_ Adsorption Performance

To mimic the post-combustion CO_2_ capture, the CO_2_ adsorption performances of the materials were detected from a simulated flue gas (a mixture of CO_2_ and N_2_ with 15 vol.% CO_2_) at 75 °C in a fixed-bed flow system equipped with a gas analyzer (see Figure 10a). On the whole, all samples presented a rapid adsorption within a short time and reached saturated adsorption within 300 s (Figure 10b). For PAP samples, the CO_2_ capacity was increased following the order of PBP-1 < PBP-2 < PAP-1< PAP-3 < PAP-2 < PBP-3, which was overall positively correlated with their specific surface areas (see Table 1). Amongst these samples, PBP-3 presented the maximum CO_2_ capacity (0.84 mmol·g^−1^) although its surface area was somewhat lower than those of PAP-1 and PAP-2 (see Table 2). This result was probably due to the competitive effect of physisorption and chemisorption [48]. On one hand, the low specific surface area was not in favor of physisorption. On the other hand, however, chemisorption was gradually enhanced caused by the relatively high content of N-site rich motif DPTH, which could counteract the decrease in physisorption.

To further improve the CO_2_ performance of the dynamic covalent samples, amino-functional molecule tetraethylenepentamine (TEPA) was further introduced on PBP. It can be noted that the direct assembly of TEPA and DTPH failed to afford any solid powder. By virtue of the dynamic nature of imine assembly, TEPA can be facilely grafted on PBP or even replace DPTH through imine exchange [49,50]. Benefiting from the many more active sites in TEPA than that in DPTH, as expected, the CO_2_ capacity of the TEPA-modified PBP-3 sample (named TEPA-PBP) could reach up to 1.27 mmol·g^−1^ (Figure 10b), 1.5 times that of bulk PBP-3. The high adsorption capacity for TEPA-PBP was mainly caused by the porous property and good affinity of amino/acylamide groups. The amino groups could act as chemisorbed sites to react with CO_2_ to form ammonium carbamate [51], and acylamide and imine groups could adsorb CO_2_ under the effect of hydrogen bonding and electrostatic attraction, respectively (see Figure 3) [51].

The cyclic stability was also crucial for the industrial application of an adsorbent. The cyclic stability of TEPA-PBP was investigated by performing 10 cycles of CO_2_ adsorption at 75 °C. It can be seen that the CO_2_ adsorption capacity of TEPA-PBP remained constant in each cycle (Figure 10c), reaching 1.21 mmol·g^−1^ of CO_2_ uptake after 10 cycles, verifying the excellent adsorption reversibility of TEPA-PBP.

The CO_2_ adsorption performance of the designed material TEPA-PBP was compared with typical reported adsorption materials. As shown in Table 3, TEPA-PBP exhibited competitive CO_2_ adsorption capacity from flue gas. Superior to most reported adsorbents, moreover, the material developed in this work exhibits facile preparation, low cost, and a metal-free nature, thus promoting its availability in practical application.

## 3. Materials and Methods

### 3.1. Materials

Dimethyl 3,3′-dithiobispropionate (98%) and tetraethylenepentamine (98%) were purchased from Shanghai Titan Technology Co., Ltd., Shanghai, China. *p*-Phthalaldehyde (PA, 98%) and 4,4′-biphenyldicarboxaldehyde (BPDA, 98%) were purchased from Innochem Chemical Co., Beijing, China. CO_2_ (99.99%) and N_2_ (99.99%) were purchased from Taiyuan Steel Co., Shanxi, China.

### 3.2. Synthesis of Materials

#### 3.2.1. Synthesis of 3,3′-Dithiobis(Propionyl Hydrazine)

Dimethyl 3,3′-dithiodipropionate (12.01 g, 50.4 mmol) was added to anhydrous methanol (90 mL), followed by the addition of hydrazine hydrate (20.59 g, 403.2 mmol). After stirring at room temperature for 24 h, a white solid (3,3′-dithiobis(propionyl hydrazine), DTPH) was obtained by centrifugation, washed with methanol (30 mL × 2) and diethyl ether (30 mL × 2), and then dried in vacuum at 50 °C for 12 h. ^1^H-NMR (δ_ppm_, 400 MHz, DMSO-d6): 9.09 (s, 1H), 4.20 (m, 2H), 2.88 (m, 2H), 2.40 (m, 2H).

#### 3.2.2. Synthesis of PAP-X

PA (0.78 g, 3.3 mmol) was dissolved in 15 mL of ethyl acetate at 30 °C, and the solution was slowly added to 15 mL of DTP aqueous solution (0.45 g, 3.3 mmol). The two-phase system was sealed and left standing at 30 °C for 24 h. White powder was collected by centrifugation, washed with water (30 mL × 3) and ethyl acetate (30 mL × 3), and then dried in vacuum at 70 °C for 12 h. The samples with different molar ratios of PA/DTP (i.e., 1:0.5, 1:1, and 1:2) were prepared and named PAP-1, PAP-2, and PAP-3, respectively.

#### 3.2.3. Synthesis of PBP-X

PBP-1 was synthesized following a similar procedure with PAP-1 under the reaction of BPDA and DTP at a molar ratio of 1:0.5. The samples with different molar ratios of BPDA/DTP (i.e., 1:0.5, 1:1, and 1:2) were prepared and named PBP-1, PBP-2, and PBP-3, respectively.

#### 3.2.4. Synthesis of TEPA-PBP

Tetraethylenepentamine (TEPA, 0.03 g, 158 mmol) was dissolved in 15 mL of methanol and stirred at room temperature for 2 h. Then, 0.50 g of PBP-3 was added to the above solution under stirring at room temperature for 8 h. The precipitate was collected by centrifugation, washed with methanol (5 mL × 2) and water (5 mL × 2) in turn, and dried at 60 °C for 12 h.

### 3.3. Characterizations

The microscopic structures of samples were observed by scanning electron microscopy (SEM) on a ZEISS MERLIN CoMPact (Zeiss, Jena, Germany). Fourier-transform infrared (FT-IR) spectra were recorded on a Tensor 27 spectrometer (Bruker, Billerica, MA, USA) over a KBr pellet in the region of 4000–400 cm^−1^. X-ray diffraction (XRD) patterns were acquired in the range of 2θ = 10°–80° with scanning rate of 5°/min on PANayltical Empyrean (Almelo, The Netherlands). Thermogravimetry analysis (TGA) was conducted using an STA 449 F3 Jupiter^®^ instrument (Netzsch, Selb, Germany) at a heating rate of 10 °C/min in N_2_ atmosphere. The specific surface areas and pore structures of samples were measured on a Brunauer-Emmett-Teller (BET) apparatus (JW-BK200, Beijing, China). The micropore size distribution was calculated using the t-plot method, the mesopore size distribution was calculated using the BJH method, and the specific surface area was analyzed using the BET equation.

### 3.4. CO_2_ Adsorption Capacity Evaluation

The CO_2_ adsorption performances were determined in a fixed-bed flow system equipped with a gas analyzer (Gasboard-3100, Wuhan Cubic Optoelectronics Co., Ltd., Wuhan, China.). Typically, 0.5 g samples were placed in a column with diameter of 0.8 cm and heated to 100 °C in N_2_ (99.999%) at a flow rate of 100 mL·min^−1^ for 1 h. Then, the column was cooled to 75 °C, and a gas mixture of CO_2_ (15 vol.%) and N_2_ flowed through the column. The total pressure was 1 bar, and the CO_2_ pressure was 0.15 bar. The CO_2_ concentration was recorded every 1.0 s until returning to 15 vol.%. After adsorption, the column was heated to 120 °C for 1 h in N_2_ (99.999%) at a flow rate of 100 mL·min^−1^ to achieve CO_2_ desorption. The CO_2_ adsorption capacity (*q_m_*, mmol·g^−1^) of the samples at a certain time (t, s) was calculated using Equation (1) [57].
(1)$$q_{m}=\frac{Q}{M_{ad}}\int_{0}^{t}\;(C_{0}-C_{t})\;dt\times \frac{T_{0}}{T}\times \frac{1}{V_{m}}$$
where *Q* is the flow of the mixture (mL·min^−1^), *M_ad_* is the mass of adsorbents, 𝑡 is the adsorption time (s), *C*_0_ and *C_t_* are the CO_2_ concentrations at the inlet and outlet, *T* is the adsorption temperature (K), and *V_m_* is the standard molar volume (22.4 L·mol^−1^, *T*_0_ = 0 °C).

## 4. Conclusions

Through dynamic covalent assembly, *N*-site rich polymers were successfully constructed via the self-assembly of 3,3′-dithiobis (propionyl hydrazine) and polyaldehyde at the ethyl acetate/water interface at 30 °C. By regulating the type and ratio of polyaldehyde, nanospheres and hollow nanotubes with spongy pores were controllably formed. The dynamic covalent materials presented good thermal stability with a maximum decomposition temperature of ca. 292 °C. Tetraethylenepentamine with rich amino-functional groups was successfully grafted onto PBP at room temperature and provided rich chemisorption sites, such that the functional dynamic covalent polymer had a high CO_2_ capacity (1.27 mmol·g^−1^) in flue gas at 75 °C. This strategy has the advantages of a simple, green, and efficient preparation process, and it provides a new idea for the controllable design of highly active CO_2_ adsorbent.

## Data Availability

Not applicable.

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
