# Peer review of "Controllable Construction of Amino-Functionalized Dynamic Covalent Porous Polymers for High-Efficiency CO2 Capture from Flue Gas"

_molecules, 2022, doi:10.3390/molecules27185853_

Round 1

Reviewer 1 Report

Comments and Suggestions for Authors

After reading the manuscript entitled "Controllable construction of amino-functionalized supramolecular porous polymers for high-efficiency CO2 capture from flue gas" carefully, the manuscript have been written with good language and I recommend it's very appropriate for publishing in Molecules Journal after make some major comments:

1.     Re-write the scheme with adding more details such as solvent, time of reaction, catalyst …

2.     Separate IR chart and SEM images each technique in different figure with clear caption.

3.     About SEM images: put all image in one magnification.

4.     I think there is no need to Vmicro. So, delete this column.

5.     The authors mention XRD and TGA techniques in 3.3. characterization but there is no figures, no results, no discussion.

6.     What about the pressure that have been used in CO2 storage?

7.     Why the authors choose the CO2 gas?

8.     Add table contain CO2 capacity, wt%.

9.     Kindly, add clear explanation. Why PAP-2 and PBP-3 have higher surface area?

10.  Add below references:

Ahmed, D. S., El-Hiti, G. A., Yousif, E., Hameed, A. S., & Abdalla, M. (2017). New eco-friendly phosphorus organic polymers as gas storage media. Polymers9(8), 336.

Hadi, A. G., Jawad, K., Yousif, E., El-Hiti, G. A., Alotaibi, M. H., & Ahmed, D. S. (2019). Synthesis of telmisartan organotin (IV) complexes and their use as carbon dioxide capture media. Molecules24(8), 1631.

Omer, R. M., Al-Tikrity, E. T., El-Hiti, G. A., Alotibi, M. F., Ahmed, D. S., & Yousif, E. (2019). Porous aromatic melamine Schiff bases as highly efficient media for carbon dioxide storage. Processes8(1), 17.

Reviewer 2 Report

  The idea is good but the introduction lacks proper justification of the research gap.

- The kinetics, isotherms, and thermodynamics data are missing.

- Abstract and conclusions are almost the same.

- It is unclear which equipment was used to determine the concentration of CO2.

- It has been claimed that TGA and XRD analysis was also conducted but no results of such characterizations are provided.

-  I reject the paper and suggest authors resubmit it after extensive revision of all sections specifically the results.

Reviewer 3 Report

Yi and coworkers have synthesized a series of amino-functional porous materials through the Schiff base reaction. The materials exhibited high-efficiency CO2 capture ability from its mixture with N2, which may provide a feasible way for capturing CO2 from flue gas. I recommend the publication of this work in Molecules, but several comments should be considered.

(1) The authors claimed the reaction between the aldehyde and amino groups is an assembly process. However, this process favors covalent synthesis rather than supramolecular assembly.

(2) The authors should clarify the mechanism for the uptake of CO2 by the porous materials. Please detail discuss the interaction mode between the amino group and CO2.

(3) The authors should provide more characterizations to figure out the constitution of the resulting polymers. For example, solid-state 13C NMR, elemental analysis, and others.

(4) How about the performance of the polymer directly obtained from tetraethylenepentamine (TEPA) and 3,3'-dithiobis (propionyl hydrazine) (DTPH)?

Round 2

Reviewer 2 Report

I accept the paper.

Reviewer 3 Report

My comments are treated appropriately, and I recommend accepting the current version.